🔓 PLOS | ONE

# Virus-associated anterior uveitis and secondary glaucoma: Diagnostics, clinical characteristics, and surgical options

**Dominika Pohlmann**[1‡]*, **Milena Pahlitzsch**[1‡]*, **Stephan Schlickeiser**[2], **Sylvia Metzner**[1], **Matthias Lenglinger**[1], **Eckart Bertelmann**[1], **Anna-Karina B. Maier**[1], **Sibylle Winterhalter**[1], **Uwe Pleyer**[1]

**1** Berlin Institute of Health, Charité–Universitätsmedizin Berlin, corporate member of Freie Universität Berlin, Humboldt-Universität zu Berlin, Berlin, Germany, **2** Institute of Medical Immunology, Charité–University Medicine, Berlin, Berlin, Germany

‡ These authors share first authorship on this work.
* dominika.pohlmann@charite.de (DP); milena.pahlitzsch@charite.de (MP)

**Data Availability Statement:** All relevant data are within the Supporting Information files.

## Abstract

In this retrospective, single-center, observational study, we compared the clinical characteristics, analyzed the glaucoma development, and the glaucoma surgery requirement mediators in patients with different virus-associated anterior uveitis (VAU). In total, 270 patients (= eyes) with VAU confirmed by positive Goldmann-Witmer coefficients (GWC) for cytomegalovirus (CMV), herpes simplex virus (HSV), varicella-zoster virus (VZV), rubella virus (RV), and multiple virus (MV) were included. Clinical records of these patients were analyzed. Demographic constitution, clinical findings, glaucoma development, and surgeries were recorded. The concentrations of 27 immune mediators were measured in 150 samples of aqueous humor. The GWC analysis demonstrated positive results for CMV in 57 (21%), HSV in 77 (29%), VZV in 45 (17%), RV in 77 (29%), and MV in 14 (5%) patients. CMV and RV AU occurred predominantly in younger and male patients, while VZV and HSV AU appeared mainly with the elderly and females (P<0.0001). The clinical features of all viruses revealed many similarities. In total, 52 patients (19%) showed glaucomatous damage and of these, 27 patients (10%) needed a glaucoma surgery. Minimal-invasive glaucoma surgery (MIGS) showed a reliable IOP reduction in the short-term period. In 10 patients (37%), the first surgical intervention failed and a follow-up surgery was required. We conclude that different virus entities in anterior uveitis present specific risks for the development of glaucoma as well as necessary surgery. MIGS can be suggested as first-line-treatment in individual cases, however, the device needs to be carefully chosen by experienced specialists based on the individual needs of the patient. Filtrating glaucoma surgery can be recommended in VAU as an effective therapy to reduce the IOP over a longer period of time.

**Funding:** Dr. Dominika Pohlmann is participant in the BIH Charité Clinician Scientist Program funded by the Charité- Universitätsmedizin Berlin and the Berlin Institute of Health. We acknowledge support from the German Research Foundation (DFG) and the Open Access Publication Fund of Charité – Universitätsmedizin Berlin.

**Competing interests:** The authors have declared that no competing interests exist.

## Introduction

Virus-associated anterior uveitis (VAU) is caused by cytomegalovirus (CMV), herpes simplex virus (HSV), varicella-zoster virus (VZV), and rubella virus (RV). The most common VAU is the herpetic cause which includes CMV, HSV, and VZV and accounts for 5% to 10% of all uveitis cases seen at tertiary referral centers.[1–4] In most cases, the diagnosis is made based on clinical characteristics. Even though the individual viruses may show some subtle differences, there are several overlapping signs which make the diagnosis challenging. Although the aqueous humor (AH) can be easily examined by polymerase chain reaction (PCR) or Goldmann-Witmer coefficient (GWC), the confirmation of the virus in AH is often not done. Additionally, prior studies reported that reverse transcription (RT)—PCR has frequently failed to demonstrate the presence of RV RNA in Fuchs'Uveitis Syndrome (FUS) due to a low-viral load below detection level and a high-rate of anti-RV antibodies which block the viral replication. [5–9] For the analysis of immunoglobulin fraction in the AH and serum, the use of GWC is mandatory to differentiate RV, but it is also mandatory for all other herpetic viruses especially in a period of latency. Previous results showed that immune mediators play a crucial role in specific viral inflammation and influence intraocular pressure (IOP). CMV demonstrated a stronger active inflammatory response, while RV may trigger chronic inflammation.[10] Inflammatory effects on the IOP levels differ between the virus types. CMV is a well-known cause of secondary glaucoma with high IOP levels >30mmHg,[11] first described by Posner and Schlossman in 1948.[12–14] Approximately 10–40% of VAU patients could develop glaucoma.[15–20] One very important risk factor for the development of chronic glaucoma is the number of IOP peaks.[15] Additionally, patients developing glaucoma usually present themselves with high IOP levels at their first inflammatory episode.[15] The amount of the viral load is also significantly associated with the number of uveitic recurrences.[21] Finally, 19% of VAU patients (VZV and HSV) needed surgical intervention to control individually elevated IOP levels.[15]

In this study, we examined 270 patients (= eyes) with VAU of which 52 developed secondary glaucoma. The diagnosis was made based on the detection of CMV, HSV, VZV, and RV to compare their demography and clinical characteristics. We place an emphasis on the IOP development and glaucoma therapy, considering glaucoma medication and different surgical therapeutic approaches. In addition, we measured immune mediators in AH of 150 eyes to add more rigor to the clinical findings.

## Methods

The retrospective, single-center study design complies with the ethical principles for medical research as outlined in the Declaration of Helsinki approved by the local ethics committee (EA 4/054/16) of Charité University Medicine Berlin. From January 2009 to December 2018, a total of 270 immunocompetent patients (= eyes) with CMV (57), HSV (77), VZV (45), RV (77), and multiple virus (MV) (14) were included. For routine diagnostic purpose, AH samples were obtained from all patients to analyze the antibody synthesis by GWC as described previously.[10] Patients with more than one positive virus of the aforementioned viruses were summarized in the group MV. The patients gave informed consent before anterior chamber (AC) stab incision and glaucoma surgery. Data were fully anonymized before analysis.

The following clinical characteristics were collected from patient´s medical records before the AC stab incision and in patients who underwent glaucoma surgery pre- and postoperatively: unilateral or bilateral uveitis, acute or chronic course, previous keratitis, visual acuity (VA) in log of the Minimum Angle (logMAR), IOP, conjunctival redness, corneal edema, keratic precipitates (KP), character of KPs, location of KPs, cells in AC, hypopyon, fibrin, iris

synechia, irisatrophy, heterochromia, lens status, vitreous involvement, and macular edema. Inflammation was evaluated using scoring criteria set out by the Standardization of Uveitis Nomenclature (SUN) working group.[22] Macular edema was detected by optical coherence tomography (SPECTRALIS® Heidelberg Engineering, Heidelberg, Germany). Additionally, glaucoma therapy, as well as the systemic antiviral therapy were documented. Apart from the topical steroids, patients with HSV received oral acyclovir (400mg / 5x day for 4–6 weeks in acute course; 400mg / 3x day as maintenance dose) and VZV patients were treated with a higher dose of acyclovir (800mg / 5x day for 4–6 weeks; 400mg / 3x day as maintenance dose). In cases of side-effects or non-response, the therapy was switched to oral valacyclovir (HSV: 500 to 1000mg twice a day; VZV: 1000mg 3x day). Patients with CMV received 900 mg valgan-cyclovir twice a day for two weeks; 450mg twice a day for 3–6 months. Elevated IOP >21mmHg was treated with topical anti-glaucomatous therapy and IOP >30mmHg addition-ally with oral acetazolamide.

## Goldmann-Witmer coefficient

The AH samples were immediately processed after the AC stab incision. A modified ELISA technique (Enzynost®, Dade Behring Marburg, Germany) was performed to detect antibodies in AH and serum, diluted to an IgG level of 1 mg/dL after total IgG in the serum and AH were measured.[6,23] A comparison of photometric signals of $\Delta E > 0.2$ allowed for detection of intraocular IgG antibodies to CMV, HSV, VZV, and RV. The antibody index (AI) was deter-mined using the GWC.[23,24] The diagnosis was confirmed by AI > 3.0 or $\Delta E > 0.200$ for all viruses.

## Immune mediator analysis

From 270 AH samples, we were able to measure the concentration of immune mediators in 150 AH samples by Bio-Plex Pro™ magnetic color-bead-based multiplex assay (Bio-Rad Lab-oratories, Inc. Hercules, CA). Fifty microliters of AH were used for the measurement. Samples with insufficient material could not be measured. The following 27 immune mediators were analyzed: eotaxin, fibroblast growth factor basic (FGFbasic), granulocyte-colony stimulating factor (G-CSF), granulocyte macrophage colony-stimulating factor (GM-CSF), interleukin-1 receptor antagonist (IL-1RA), IL-1b, IL-2, IL-4, IL-5, IL-6, IL-7, IL-8, IL-9, IL-10, IL-12, IL-13, IL-15, IL-17, interferon-gamma (IFN-γ), interferon gamma-induced protein 10 (IP-10), mac-rophage inflammatory proteins 1 alpha and beta (MIP-1α and MIP-1β), monocyte chemotac-tic protein 1 (MCP1), platelet-derived growth factor (PDGF), regulated upon activation normal T cell expressed and secreted (RANTES), tumor-necrosis-factor-alpha (TNF-α), and vascular endothelial growth factor (VEGF). The assay was conducted according to manufac-turer's instruction. Data analysis was performed by Bio-Plex Manager ™ software 1.1.

## Glaucoma criteria

Glaucoma was defined by an optic neuropathy showing characteristic glaucomatous optic disc alterations and visual field defects. A major risk factor was IOP elevation. All patients included in this study demonstrated an open chamber angle in the gonioscopy (Shaffer III-IV, well-pig-mented trabecular meshwork, no neovascularization). The optic disc was classified using the diagnostic criteria described by Jonas.[25] The inclusion criteria entailed best-corrected visual acuity of at least 20/200 and informed patient consent for surgery, if necessary. The intraocular pressure (IOP) was measured by using the well-known Goldmann applanation tonometry.[26] Preoperatively and in the subsequent visits post-surgery IOP readings, VA in logMAR and the number of glaucoma medications were analyzed. Success rate of therapy is defined by a

controlled IOP and inflammatory situation which might differ in individuals due to their severity of glaucomatous damage. Failure rate was defined by a second intervention due to uncontrolled individual pressure in secondary glaucoma. Patients attended clinics preoperatively as well as follow-ups one day, six weeks, three months, six months, one year, and two years postoperatively.

## Glaucoma surgery

First line surgical therapy was minimal-invasive glaucoma surgery (MIGS) including Trabectome® surgery and iStent inject® implantation. Surgical interventions were always conducted under stable inflammatory conditions. Furthermore, we considered the canaloplasty as a first line procedure (MIGS) to reduce IOP. Two patients received cyclophotocoagulation (inferior hemisphere, 20 spots, 2000 mW, 2000 mseconds) in our retrospective data set. Second line therapy involved filtrating glaucoma surgery such as trabeculectomy. The surgeries were conducted by two surgeons (SW, EB).

## Minimal-invasive glaucoma surgery

Surgical scheme of MIGS in short[27–30]: a 1.8 mm incision in the limbal temporal cornea was made, acetylcholine chloride 1% (Miochol) was inserted individually as needed and followed by an injection of ophthalmic viscosurgical solution to contain the AC morphology. At this point, there are two options. First option–the Trabectome® (NeoMedix, Inc., Tustin, CA, USA) handpiece was inserted and the selective electrosurgical ablation was activated to remove an 120˚ arc of trabecular meshwork and an inner wall of the Schlemm canal.[27,30] Alternatively, iStent inject® (Glaucos Corporation, Laguna Hills, CA, USA) implantation was chosen and two iStents were implanted through the nasal trabecular meshwork into Schlemm's canal, usually separated by two hours.[28,29] MIGS were conducted under gonioscopic view.

## Canaloplasty ab externo

Lewis et al. published this surgical procedure in detail.[31] In brief, after conjunctival limbal opening at the upper quadrant, a non-penetrating two-flap dissection of the sclera was prepared to expose Schlemm's canal.[31] The iTrack-microcatheter (Ellex iScience Inc., Fremont, CA, USA) was used to dilate the full circumference of the canal with the assistance of sodium hyaluronate (Healon GV, Advanced Medical Optics, Inc., Santa Ana, CA, USA). Catheterization was conducted over the complete circumference. A 10-0 Prolene suture (Ethicon, Inc., Somerville, NJ, USA) was then applied to the microcatheter tip and left in the canal with both ends tightened to expand the trabecular meshwork inward.[31,32]

## Filtrating surgery

Trabeculectomy can be used as first or second line therapy to reduce high-levels of uncontrolled IOP with use of Mitomycin C (0.2 mg/ml). In essence, a fornix-based peritomy of conjunctiva in the upper quadrant was created and an approximately 2.5 × 2.5 mm scleral flap was dissected, [33–35] followed by a Mitomycin C (0.2 mg/ml) sponge application on the scleral surface for three minutes before lavage with generous balanced salt solution (BSS). Intralamellar scleral sutures using 10–0 nylon (Alcon, Camberley, UK) were pre-placed at the corners of the scleral flap, and a paracentesis was placed temporarily. Furthermore, a sclerostomy (500µm) was created with a Khaw Descemet membrane punch 7–101 (Duckworth & Kent, Baldock, UK) and a surgical iridectomy was conducted. Finally, the pre-placed sutures were tightened, and the conjunctival tissue closed.[33–35]

## Statistical analysis

Data of clinical characteristics and cytokine concentrations were analyzed by using GraphPad Prism 8 (GraphPad Software, La Jolla, CA) and SPSS (Version 20.0). For demographic and clinical parameters descriptive statistics (mean, standard deviation), Chi-Square and Fisher test were performed. For differences in cytokine concentrations, non-parametric Mann-Whitney testing was performed. Two-tailed, non-parametric Spearman method was applied to assess the correlation between variables. Comparison of preoperative to postoperative glaucoma parameters was conducted by the independent sample t-test. Numeric variables, which do not show a normal distribution range, were compared with the Mann–Whitney U test (two cohorts) and the Kruskal-Wallis test (three cohorts). Spearman´s correlation was analyzed to correlate AI coefficient of all virus types and IOP. For testing normality, the Kolmogorov-Smirnov-Test was applied. A p-value of <0.05 indicated a statistically significant difference.

# Results

## Demography

The GWC analysis demonstrated positive results for CMV in 57 (21%), for HSV in 77 (29%), for VZV in 45 (17%), for RV in 77 (29%), and MV (8 VZV+HSV; 2 VZV+CMV; 2 HSV +CMV; 1 VZV+RV; 1 CMV+RV) in 14 (5%) out of 270 patients (= eyes) (Fig 1). The CMV (median age, 42; range 19–89) and RV (median age, 44 years; range 19–76) patients were younger than the HSV (median age, 56; range 19–87), VZV (median age, 67; range 19–96), and especially the MV cohort (median age, 75; range 28–81) (P <0.0001). The male and female ratio did not differ between CMV, HSV, VZV, RV, and MV patients (P = 0.1029). Unilateral involvement was typical for almost all viruses (76–98%), but one quarter of VZV (24%) patients also had a bilateral manifestation (P = 0.0149). To note, the second eye presented only a keratitis, not a keratouveitis, The course of disease was predominantly acute for CMV (55/57; 96%) patients, while chronic disease was observed in RV (52/77; 84%) and in HSV patients (52/77; 68%) (P<0.0001). VZV and MV patients equally demonstrated an acute and a chronic course of disease. Almost half of the patients (150/270; 56%) revealed a previous keratitis, in particular HSV (44/77; 57%) and VZV (23/45; 51%) patients (P<0.0001). No VZV patient had a history of previous herpes zoster ophthalmicus (no dermal lesions). The worst VA revealed VZV patients with 0.5 logMAR, followed by MV with 0.3 logMAR, and HSV/RV patients with 0.2 logMAR (P>0.0001). The CMV patients had the best VA with 0 logMAR, although they presented the highest median IOP with 27 mmHg compared to VZV (18 mmHg), HSV, RV (16 mmHg), and MV (16.5 mmHg) (P<0.0001). The CMV patients also possessed the highest number of local and systemic anti-glaucomatous drugs compared to other patients (P<0.0001). More details of demographic data are presented in Table 1.

## Clinical findings

The ophthalmologic findings from the preoperative day are found in Table 2. Conjunctival redness was presented more often in CMV (29/57; 54%) and VZV (21/45; 47%) in comparison to RV (21/77; 27%) (P = 0.0086; P = 0.0435). Corneal edema was observed only in a few patients (1–7%). The occurrence of KPs differed between the groups (P = 0.04). HSV (54/77; 70%) and VZV (30/45; 67%) patients showed significant differences compared to RV (67/77; 87%) patients. The character and location of KPs were not reported consistently and therefore could not be analyzed accurately. Iris synechia were especially noted in HSV (12/77; 12%) patients compared to CMV (0/57; 0%) patients (P = 0.0352), while iris atrophy was not noted in a specific patient group (P = 0.3293). Anterior inflammation was observed in all patients

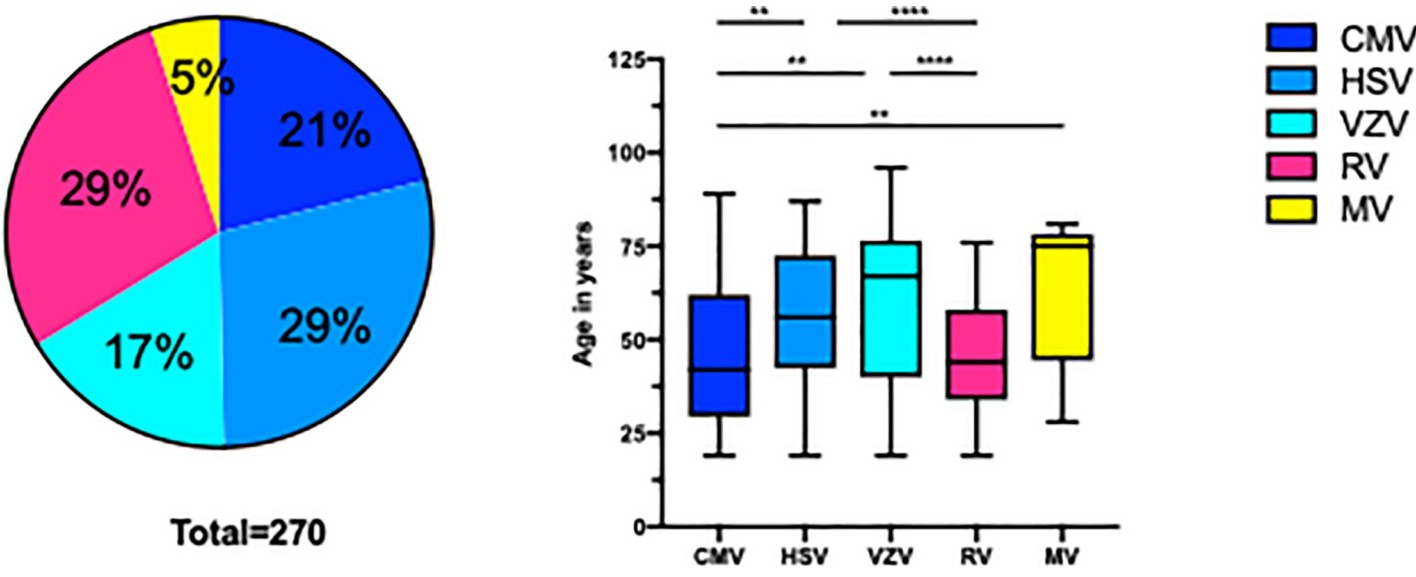

**Fig 1. Virus distribution of the complete study cohort and age association of virus infection.**

(P = 0.5991), but VZV patients showed more severe inflammation with hypopyon (2/45; 4%), vitreous haze (8/45; 31%), and cystoid macula edema (CME) (5/45; 11%). In contrast, iris synechia were not present, but iris heterochromia (46/77; 60%) (P<0.0001), cataract at presentation (55/77; 71%), and vitreous haze occurred more frequently (47/77;61%) (P<0.0001) in RV patients.

## Glaucoma

In total, 52 patients (= eyes) out of 270 (19%) (mean age 48.0±18.8 years) with VAU, showed glaucomatous damage in accordance to Jonas criteria[15] and were diagnosed with secondary glaucoma (see Table 3 for descriptive statistics). Of these, a total of 27 patients (= eyes) needed glaucoma surgery (10% of all VAU) (RV: 11; CMV: 6; HSV: 5, VZV: 3; MV: 2). In detail, 16 patients (62.8±12.4years) received MIGS including iStent inject® and Trabectome®, in 2 patients (52.5±31.8years) cyclophotocoagulation was performed, and 9 patients (52.1±19.7 years) obtained trabeculectomy and Mitomycin C. Nine VAU patients had a trabeculectomy as first line procedure due to unavailability and missing experience of newly developed MIGS devices in uveitis at that point in time (retrospective data from 01/2009 onwards). Results are outlined in Table 4. IOP preoperatively was reduced from 28.18±9.32mmHg to 14.72 ±6.67mmHg two years postoperatively (P = 0.004). In addition, a significant reduction of glaucoma medication can be reported from preoperatively 1.91±0.94 to 0.81±1.16 two years after surgery (P = 0.040). The trendline chart of IOP and glaucoma medication can be found in Fig 2. Follow-up interventions in individually uncontrolled IOP were carried out in 10 eyes (37%) (IOP >16 mmHg) (Fig 3). The first surgical intervention failed in nine patients receiving MIGS (Trabectome®: 4; iStent inject®: 3; Canaloplasties ab externo: 2) and failed in one patient receiving trabeculectomy and Mitomycin C. No patient with MV needed further surgeries. MIGS did not show any perioperative complications other than blood reflux. Blood reflux can be used as a parameter to assess successful trabecular meshwork surgery. Furthermore, there were no incidences of sustained hypotony, choroidal effusion, hemorrhage, infection, aqueous misdirection, or wound leakage in all surgeries that were carried out. Duration until performed follow-up interventions is outlined in Fig 3.

**Table 1. Demographic parameters of all virus-associated anterior uveitis cohorts.**

| | Total | CMV | HSV | VZV | RV | MV | P Value | | | | | | | | | | |
|---|---|---|---|---|---|---|---|---|---|---|---|---|---|---|---|---|---|
| | | | | | | | CMV vs HSV | CMV vs VZV | CMV vs RV | CMV vs MV | HSV vs VZV | HSV vs RV | HSV vs MV | VZV vs RV | VZV vs MV | RV vs. MV |
| Number of patients (%) | 270 | 57 (21) | 77 (29) | 45 (17) | 77 (29) | 14 (5) | | | | | | | | | | |
| Median age at presentation (range) | 52 (19–96) | 42 (19–89) | 56 (19–87) | 67 (19–96) | 44 (19–76) | 75 (28–81) | <0.0001 | 0.0022 | 0.0018 | 0.7373 | 0.0021 | 0.3242 | 0.0004 | 0.1183 | 0.0004 | 0.3465 | 0.0004 |
| Gender | | | | | | | | | | | | | | | | |
| Female (%) | 120 (44) | 21 (37) | 40 (52) | 25 (56) | 30 (39) | 4 (29) | 0.1029 | 0.0826 | 0.0593 | 0.8028 | 0.5616 | 0.3853 | 0.1056 | 0.1074 | 0.0755 | 0.0778 | 0.4598 |
| Male (%) | 150 (56) | 36 (63) | 37 (48) | 20 (44) | 47 (61) | 10 (71) | | | | | | | | | | |
| Median of Antibody-Index (GWC) (range) | 5.6 (1.31–485.5) | 7.265 (1.6–485.5) | 5.495 (2.1–109.7) | 4.330 (1.31–106.6) | 6.450 (1.89–199.7) | 3.94 (1.91–83.33) | 0.0038 | 0.0423 | 0.0047 | 0.6101 | 0.0021 | 0.1842 | 0.0972 | 0.0708 | 0.0059 | 0.6107 | 0.0028 |
| Unilateral | 230 (85) | 56 (98) | 66 (86) | 34 (76) | 62 (81) | 12 (86) | 0.0149 | 0.012 | 0.0004 | 0.0018 | 0.0368 | 0.1591 | 0.3895 | >0.999 | 0.5183 | 0.4232 | 0.6464 |
| Bilateral | 40 (15) | 1 (2) | 11 (14) | 11 (24) | 15 (19) | 2 (14) | | | | | | | | | | |
| Acute course | 123 (46) | 55 (96) | 25 (32) | 23 (55) | 10 (16) | 7 (50) | <0.0001 | <0.0001 | <0.0001 | <0.0001 | <0.0001 | 0.042 | 0.0274 | 0.2063 | <0.0001 | 0.9421 | 0.006 |
| Chronic course | 147 (54) | 2 (4) | 52 (68) | 22 (49) | 52 (84) | 7 (50) | | | | | | | | | | |
| Previous keratitis | 150 (56) | 0 (0) | 44 (57) | 23 (51) | 25 (40) | 6 (43) | <0.0001 | <0.0001 | <0.0001 | <0.0001 | <0.0001 | 0.5183 | 0.0487 | 0.320 | 0.268 | 0.5895 | 0.816 |
| Visual acuity median in logMAR (range) | 0.2 (0–2.5) | 0 (0–2) | 0.2 (-0.1–2.5) | 0.5 (0–2.5) | 0.2 (0–2.5) | 0.3 (0–1) | <0.0001 | 0.0028 | <0.0001 | 0.0020 | 0.0035 | 0.0028 | 0.8261 | 0.5886 | 0.0003 | 0.0520 | 0.3528 |
| Intra ocular pressure (IOP) median in mmHG (range) | 18 (6–58) | 27 (10–55) | 16 (6–38) | 18 (7–40) | 16 (8–58) | 16.5 (8–42) | <0.0001 | <0.0001 | 0.0009 | <0.0001 | 0.0163 | 0.1116 | 0.8376 | 0.7248 | 0.1571 | 0.7274 | 0.7913 |
| IOP | | | | | | | | | | | | | | | | |
| >21–29 mmHg | 81 (30) | 27 (47) | 15 (19) | 14 (31) | 22 (29) | 3 (21) | <0.0001 | 0.0002 | 0.2419 | 0.0007 | 0.4906 | 0.0099 | 0.7416 | 0.0037 | 0.0339 | 0.0632 | 0.0104 |
| Acute course | 48 (59) | 27 (100) | 8 (53) | 6 (43) | 5 (23) | 2 (67) | | | | | | | | | | |
| Chronic course | 33 (41) | 0 (0) | 7 (47) | 8 (57) | 17 (77) | 1 (33) | | | | | | | | | | |
| IOP | | | | | | | | | | | | | | | | |
| >30 mmHg | 39 (14) | 19 (33) | 3 (4) | 8 (18) | 6 (8) | 3 (21) | 0.3718 | 0.1487 | 0.3486 | 0.4373 | 0.9286 | 0.1394 | 0.5 | 0.2 | 0.7299 | 0.2606 | 0.2976 |
| Acute course | 25 (64) | 19 (100) | 1 (33) | 3 (38) | 0 (0) | 2 (67) | | | | | | | | | | |
| Chronic course | 14 (36) | 0 (0) | 2 (67) | 5 (62) | 6 (100) | 1 (33) | | | | | | | | | | |
| Number of local antiglaucomatous eye drops | | | | | | | <0.0001 | <0.0001 | <0.0001 | <0.0001 | 0.1487 | 0.2645 | 0.3967 | 0.6740 | 0.2628 | 0.042 | 0.3138 |
| 0 | 125 | 10 | 38 | 27 | 45 | 5 | | | | | | | | | | |
| 1 | 50 | 5 | 16 | 10 | 16 | 3 | | | | | | | | | | |
| 2 | 61 | 21 | 18 | 8 | 10 | 4 | | | | | | | | | | |
| 3 | 34 | 21 | 5 | 0 | 6 | 2 | | | | | | | | | | |
| Systemic acetazolamide | | | | | | | | | | | | | | | | |
| yes | 222 (82) | 34 (60) | 69 (90) | 35 (78) | 71 (82) | 10 (71) | <0.0001 | <0.0001 | 0.052 | <0.0001 | 0.4159 | 0.0754 | 0.5606 | 0.0644 | 0.0227 | 0.6258 | 0.0222 |
| no | 48 (18) | 23 (40) | 8 (10) | 10 (22) | 6 (8) | 4 (29) | | | | | | | | | | |

CMV = cytomegalovirus, GWC = Goldmann–Witmer coefficient, HSV = herpes simplex virus, IOP = Intraocular pressure, logMAR = log of the Minimum Angle,

MV = multiple virus, RV = rubella virus, VZV = varicella-zoster virus

Kruskal-Wallis test, Chi-square-Test, Mann-Whitney Test were performed.

Table 5 presents results of the AI coefficient in comparison to glaucoma parameters. We found higher AI values in RV and CMV patients in the IOP >30 mmHg cohort compared to a moderate IOP increase (RV: IOP >30 mmHg, AI 104.8±89.3 vs. IOP <30 mmHg 21.3±42.9). The results are not statistically significant but might be an interesting trend to follow. In addition, no significance was found between AI coefficient and topical/systemic glaucoma therapy (P>0.05). The parameter 'glaucoma surgery´ did not show any statistical significance to AI coefficient in all study cohorts.

## Immune mediators

In a total of 150 patients (CMV:23; HSV: 34; VZV:16; RV: 77), 27 immune mediators were measured. The median concentration of all cytokines was similar in all groups, except of six

**Table 2. Clinical findings of all virus-associated anterior uveitis cohorts.**

| | CMV N = 57 | HSV N = 77 | VZV N = 45 | RV N = 77 | MV N = 14 | P Value | CMV vs HSV | CMV vs VZV | CMV vs RV | CMV vs MV | HSV vs VZV | HSV vs RV | HSV vs MV | VZV vs RV | VZV vs MV | RV vs. MV |
|---|---|---|---|---|---|---|---|---|---|---|---|---|---|---|---|---|
| **Conjunctival redness** | | | | | | 0.0813 | 0.0929 | 0.4224 | **0.0086** | 0.5907 | 0.2627 | 0.3018 | 0.6441 | **0.0435** | 0.8027 | 0.2867 |
| no | 29 (54) | 28 (36) | 21 (47) | 21 (27) | 6 (43) | | | | | | | | | | | |
| yes | 28 (49) | 49 (64) | 24 (53) | 55 (71) | 8 (57) | | | | | | | | | | | |
| **Corneal edema** | | | | | | 0.4614 | 0.1824 | 0.4321 | 0.7663 | 0.7846 | 0.6983 | 0.0958 | 0.1701 | 0.2925 | 0.888 | 0.9282 |
| no | 54 (95) | 77 (99) | 44 (98) | 72 (94) | 13 (93) | | | | | | | | | | | |
| yes | 3 (5) | 1 (1) | 1 (2) | 5 (6) | 1 (7) | | | | | | | | | | | |
| **Keratic precipitate** | | | | | | **0.0400** | 0.2507 | 0.1627 | 0.2127 | 0.2494 | 0.6902 | **0.0107** | 0.6630 | **0.0072** | 0.8694 | **0.035** |
| no | 12 (21) | 23 (30) | 15 (33) | 10 (13) | 5 (36) | | | | | | | | | | | |
| yes | 45 (79) | 54 (70) | 30 (67) | 67 (87) | 9 (64) | | | | | | | | | | | |
| **Character of keratic precipitates** | | | | | | 0.9713 | 0.52 | 0.8824 | 0.8824 | 0.507 | 0.8858 | 0.8858 | 0.7752 | >0.9999 | 0.6360 | 0.6360 |
| not documented | 14 (25) | 41 (53) | 22 (49) | 28 (36) | 9 (64) | | | | | | | | | | | |
| documented | 16 (28) | 18 (23) | 10 (22) | 26 (34) | 3 (21) | | | | | | | | | | | |
| granulomatous | 21 (37) | 14 (18) | 10 (22) | 19 (25) | 2 (14) | | | | | | | | | | | |
| Fine | 6 | 4 (5) | 3 (7) | 4 | 0 (0) | | | | | | | | | | | |
| Pigmented | (11) | | | (5) | | | | | | | | | | | | |
| **Location of keratic precipitate** | 14 (25) | 38 (49) | 37 (82) | 40 (52) | 5 (36) | 0.6053 | 0.2713 | 0.8841 | 0.8913 | 0.3416 | 0.2798 | 0.2267 | 0.47459 | 0.3226 | 0.3226 | 0.3072 |
| not documented | 14 (25) | 11 (14) | 11 (24) | 15 (19) | 2 (14) | | | | | | | | | | | |
| Arlt´s triangle | 25 (44) | 28 (36) | 29 (64) | 22 (29) | 7 (50) | | | | | | | | | | | |
| diffuse scattered | 2 (4) | 0 (0) | | | | | | | | | | | | | | |
| endotheliitis | | 0 (0) | 0 (0) | 0 (0) | | | | | | | | | | | | |
| Cells in anterior chamber | 29 (50) | 48 (62) | 28 (62) | 28 (36) | 7 (50) | 0.5991 | 0.1846 | 0.2519 | 0.2519 | 0.9531 | 0.9899 | 0.9899 | 0.3852 | >0.999 | 0.4162 | 0.4162 |
| no | 28 (49) | 29 (38) | 17 (38) | 17 (38) | 7 (50) | | | | | | | | | | | |
| yes | 29 (50) | 48 (62) | 28 (62) | 28 (36) | 7 (50) | | | | | | | | | | | |
| Hypopyon | | | | | | 0.2434 | 0.1554 | 0.4246 | 0.2434 | 0.6177 | 0.0621 | | | 0.0621 | | |
| no | 56 (98) | 77 (100) | 43 (95) | 77 (100) | 14 (100) | | | | | | | | | | | |
| yes | 1 (2) | 0 (0) | 2 (4) | 0 (0) | 0 (0) | | | | | | | | | | | |
| Fibrin | | | | | | 0.2145 | 0.2640 | 0.5834 | 0.0898 | 0.381 | 0.1263 | 0.4976 | 0.8949 | **0.0435** | 0.2008 | 0.8096 |
| no | 53 (93) | 67 (87) | 43 (96) | 64 (83) | 12 (86) | | | | | | | | | | | |
| yes | 4 (7) | 10 (13) | 2 (4) | 13 (17) | 2 (14) | | | | | | | | | | | |
| Iris Synechiae | | | | | | **0.0043** | **0.0075** | **0.0479** | | **0.0421** | 0.3688 | **0.0285** | 0.6169 | | 0.9506 | 0.3809 |
| no | 57 (100) | 68 (88) | 42 (93) | 77 (100) | 13 (93) | | | | | | | | | | | |
| Yes | 0 (0) | 9 (12) | 3 (7) | 0 (0) | 1 (7) | | | | | | | | | | | |
| Irisatrophy | | | | | | 0.3293 | 0.3008 | 0.809 | 0.0881 | 0.1171 | 0.4711 | 0.4149 | 0.43 ns | 0.1777 | 0.2008 | 0.7839 |
| no | 55 (96) | 71 (92) | 43 (96) | 68 (88) | 12 (86) | | | | | | | | | | | |
| yes | 2 (4) | 6 (8) | 2 (4) | 9 (12) | 2 (14) | | | | | | | | | | | |
| Iris heterochromia | 57 (100) | 76 (99) | 44 (98) | 31 (40) | 14 (100) | **<0.0001** | 0.3878 | 0.258 | **<0.0001** | | 0.6983 | **<0.0001** | 0.6681 | **<0.0001** | 0.5737 | **<0.0001** |
| no | | 1 | 1 | 46 | | | | | | | | | | | | |
| yes | 0 (0) | (1) | (2) | (60) | 0 (0) | | | | | | | | | | | |
| Vitreous haze | | | | | | **<0.0001** | 0.1886 | 0.0001 | **<0.0001** | **0.0025** | **0.0041** | **<0.0001** | 0.0644 | **0.0014** | 0.857 | **0.0244** |
| no | 55 (96) | 69 (90) | 31 (69) | 30 (39) | 10 (71) | | | | | | | | | | | |
| yes | 2 (4) | 8 (10) | 8 (31) | 47 (61) | 4 (19) | | | | | | | | | | | |

(*Continued*)

**Table 2.** (Continued)

| | CMV N = 57 | HSV N = 77 | VZV N = 45 | RV N = 77 | MV N = 14 | P Value | CMV vs HSV | CMV vs VZV | CMV vs RV | CMV vs MV | HSV vs VZV | HSV vs RV | HSV vs MV | VZV vs RV | VZV vs MV | RV vs. MV |
|---|---|---|---|---|---|---|---|---|---|---|---|---|---|---|---|---|
| Macular edema | 54 | 75 | 40 | 77 | 13 | **0.0414** | 0.4208 | 0.2754 | **0.0417** | 0.7846 | 0.0511 | 0.1546 | 0.3809 | **0.0028** | 0.6679 | **0.0184** |
| no | (95) | (94) | (89) | (100) | (93) | | | | | | | | | | | |
| yes | 3 (5) | 2 (6) | 5 (11) | 0 (0) | 1 (7) | | | | | | | | | | | |
| Lens | | | | | | | | | | | | | | | | |
| Phakic | 38 | 37 | 15 | 9 | 4 | **<0.0001** | 0.154 | **0.0028** | **<0.0001** | **0.008** | 0.884 | **<0.0001** | **0.0429** | **<0.0001** | 0.6083 | **0.0002** |
| | (67) | (48) | (33) | (12) | (29) | | | | | | | | | | | |
| Corticonuclear cataract | 11 (19) | 23 (30) | 14 (31) | 55 (71) | 3 (21) | | | | | | | | | | | |
| Posterior subcapsular | 0 (0) | 3 (4) | 0 (0) | 5 (6) | 0 (0) | | | | | | | | | | | |
| cataract | 8 | 14 | 16 | 8 | 7 (50) | | | | | | | | | | | |
| Pseudophakic | (14) | (18) | (36) | (10) | | | | | | | | | | | | |

CMV = cytomegalovirus, GWC = Goldmann–Witmer coefficient, HSV = herpes simplex virus, IOP = Intraocular pressure, MV = multiple virus, RV = rubella virus, VZV = varicella-zoster virus

not documented data excluded

Kruskal-Wallis test, Chi-square-Test, Mann-Whitney Test were performed.

(S1 Table; S1 Fig). IL-12, IL-15, Eotaxin, IP-10, MCP-1, MIP-1b, and VEGF showed significant differences between CMV, HSV, and RV (P<0.005) (S1 Fig). In a further analysis, we excluded younger (<30 years) and older (>73 years) patients in each cohort to rule out the age specificity and have similar gender distribution (CMV: 16/23, 7 female/9 male; HSV: 23/34, 11 female/12 male, VZV: 7/16, 6 female /1 male, RV: 59/77, 22 female/ 37 male). Thereafter, no significant values were found in the Kruskal-Wallis-test, except of increased IP-10 in CMV compared to VZV patients (P = 0.0138). Also, Spearman correlations showed no significances between immune mediators and the age-matched group of all cohorts. However, significant differences were measured on the above-mentioned immune mediators (IL-12: P = 0.0106, IL-15: P = 0.0109, Eotaxin: P = 0.0122, IP-10: P = 0.012, MCP-1: P = 0.013, MIP-1b: P = 0.0106; VEGF: P = 0.0106) between female and male in CMV patients in the age-matched group. These results could be confirmed in the Spearman´s test (r = 0.2372; P<0.005). For all patients in the CMV cohort, significant differences of other immune mediators between female and male were also measured (IL-1RA: P = 0.0338; IL-5: 0.025; IL-9: P = 0.0146; IL-10: P = 0.0006; IL-13: 0.0088; FGFbasic: P = 0.0212; IFN-g: P00.0041; PDGF: P = 0.0043; RANTES: P = 0.267), whereby the women showed the lowest levels. Other VAU cohorts did not show any gender differences. In the CMV cohort, IL-10 (r = -0.3439, P = 0.0292), IFN-g (r = -0.448, P = 0.0321), MCP-1 (r = -0.4558, P = 0.0288), and MIP1a (r = -0.5088, P = 0.0132) were found to be negatively correlated with IOP values. Seven patients were considered treatment "naïve" meaning without receiving glaucoma treatment–which did not negatively confound the correlation since the patients were presented with low IOP values. CMV patients receiving acetazolamide (n = 10) revealed significantly lower immune mediator levels which we had already reported in our previous work.[10]

## Discussion

Our data demonstrate the epidemiology, the clinical characteristics, the development of glaucoma and surgical interventions, and the distribution of immune mediators in AH in immunocompetent patients with VAU of different entities. Out of the total 270 patients, one third were tested for HSV and one third for RV infection followed by CMV and VZV. In the

**Table 3. Demographic parameters of the glaucoma cohort.**

| | | |
|---|---|---|
| N | | 52 |
| Age (years) | | 48.0±18.8 |
| Gender (male/female) | | 36 (69.2%) / 16 (30.8%) |
| Pathogenic virus | RV | 11 (21.2%) (AI: 24.31±45.71) |
| | CMV | 32 (61.5%) (AI: 30.07±81.57) |
| | HSV | 5 (9.6%) (AI: 5.24±2.32) |
| | VZV | 3 (5.8%) (AI:3.15±0.13) |
| | MV | 1 (1.9%) (AI: 2.99) |
| Before AC stab incision | Side (right/left) | 31 (59.6%) / 21 (40.4%) |
| | Involvement (unilateral/bilateral) | 50 (96.2%) / 2 (3.8%) |
| | Course of disease (acute/chronic) | 38 (73.1%) / 14 (26.9%) |
| | Previous keratitis | 9 (17.3%) |
| | Visual acuity (logMAR) | 0.16±0.32 |
| | IOP | 29.79±10.66 mmHg |
| | Local therapy | 2.08±0.81 |
| | Systemic glaucoma therapy (acetazolamid) | 18 (34.6%) |
| | Steroid therapy (local) | 37 (71.2%) |
| Clinical findings | Conjunctival redness | 28 (53.8%) |
| | Corneal edema | 1 (1.9%) |
| | Keratic precipitates | 42 (80.8%) |
| | Character of precipitates (granulomatous/fine/pigmented /not documented/nothing) | 8 (15.4%) / 22 (42.3%) / 7 (13.5%) / 6 (11.5%) / 9 (17.3%) |
| | Location of precipitates (Arlt´s triangle/diffuse scattered/ endotheliitis/not documented) | 8 (15.4%) / 27 (51.9%) / 2 (3.8%) / 15 (28.8%) |
| | Cells in anterior chamber | 18 (34.6%) |
| | Hypopyon | 0 (0%) |
| | Fibrin | 4 (7.7%) |
| | Iris Synechiae | 2 (3.8%) |
| | Irisatrophy | 6 (11.5%) |
| | Heterochromia | 8 (15.4%) |
| | Lens (corticonuclear cataract or subcapsular posterior cataract/ pseudophacic/ clear lens | 14 (26.9%) /11 (21.2%) / 27 (51.9%) |
| | Vitreous Haze (0/+1/+2/+3) | 47 (90.4%) / 4 (7.7%) / 1 (1.9%) / 0 (0%) |
| | Retinal infiltrates | 0 (0%) |
| | Macula edema | 0 (0%) |
| Recurrence | recurrence (no /yes /not documented) | 13 (25.1%) / 37 (71.1%) / 2 (3.8%) |

AC = anterior chamber, CMV = cytomegalovirus, HSV = herpes simplex virus, MV = multiple virus, RV = rubella, VZV = varicella-zoster virus, IOP = intraocular pressure

glaucoma cohort, however, 61% of all patients had positive results for CMV, followed by RV infection (21%). Surprisingly, we measured more than two virus antibodies in 14 eyes and classified this group as MV which has not been shown yet. The range consisted of 28 years to 81 years in the MV group.

## Clinical findings of virus-associated anterior uveitis

CMV and RV AU occur predominantly in younger and male patients, while VZV and HSV AU appeared mainly in elderly patients with a predominance in females. These results are

**Table 4. Descriptive statistics of the glaucoma subgroup receiving a surgical intervention.**

| | | Glaucoma 1st surgery (n = 27) | | | Glaucoma 2nd surgery (n = 10) | |
| --- | --- | --- | --- | --- | --- | --- |
| | | MIGS (n = 16) | TE (n = 9) | CPC (n = 2) | Trabectome (n = 2) | TE (n = 8) |
| Age (years) | | 62.8±12.4 | 52.1±19.7 | 52.5±31.8 | 60.5±13.4 | 55.9±12.8 |
| Gender (male/ female) | | 16 (100%)/0 (0%) | 8 (88.9%)/ 1 (11.1%) | 2 (100%)/0 (0%) | 2 (100%)/ 0 (0%) | 8 (100%)/0 (0%) |
| Side (right/ left) | | 9 (56.2%)/7 (43.8%) | 5 (55.6%)/4 (44.4%) | 1 (50.0%)/1 (50.0%) | 0 (0%)/2 (100%) | 5 (62.5%)/3 (37.5%) |
| Virus | RV | 7 (43.8%) | 4 (44.4%) | 0 (0%) | 1 (50.0%) | 3 (37.5%) |
| | CMV | 4 (25.0%) | 2 (22.2%) | 0 (0%) | 1 (50.0%) | 2 (25.0%) |
| | HSV | 3 (18.8%) | 1 (11.1%) | 1 (50.0%) | 0 (0%) | 1 (12.5%) |
| | VZV | 1 (6.3%) | 1 (11.1%) | 1 (50.0%) | 0 (0%) | 2 (25.0%) |
| | MV | 1 (6.3%) | 1 (11.1%) | 0 (0%) | 0 (0%) | 0 (0%) |
| pre-Op VA (logMAR) | | 0.21±0.38 | 0.75±0.20 | 0.30±0.71 | 0.70±0.14 | 0.40±0.36 |
| pre-Op IOP (mmHg) | | 28.40±9.63 | 28.00±9.96 | 27.00±8.49 | 30.50±13.43 | 27.75±9.48 |
| Number of glaucoma medications | | 1.88±0.89 | 2.00±1.00 | 2.00±0.00 | 2.00 ± 0.00 | 2.25±1.04 |

CPC = cyclophotocoagulation, CMV = cytomegalovirus, HSV = herpes simplex virus, MIGS = minimal-invasive glaucoma surgery, MV = multiple virus, TE = trabeculectomy, VA = visual acuity, VZV = varicella-zoster virus

concordant to the literature.[17–20] Interestingly, studies report that CMV tends to more commonly affect Asian populations and it is not uncommon in immunocompetent patients. [36,37] Furthermore, it may present as a recurrent acute or chronic inflammation, resembling PSS, herpetic AU, or FUS in Asia.[36] Thus, Asian patients commonly present chronic CMV AU as FUS, while many studies in Europe confirmed that FUS is almost always related to RV [5,36,38] and PSS is associated with CMV[10,39,40] which is characteristically accompanied

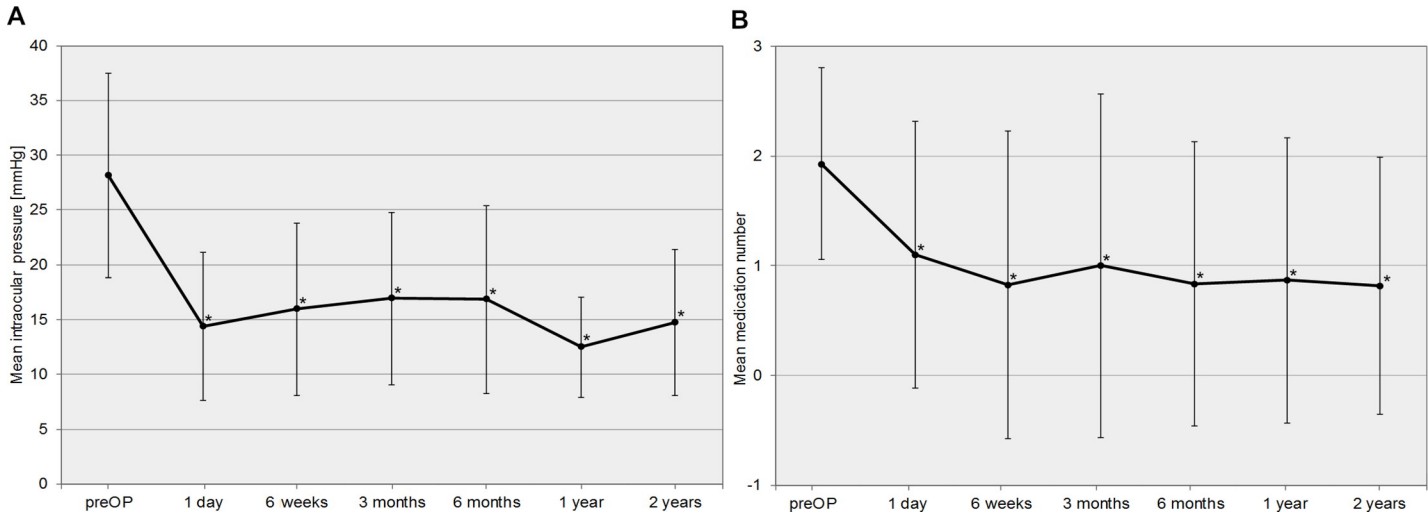

**Fig 2.** **a** Trendline chart of the intraocular pressure (IOP) comparing preoperative to postoperative follow-up data. **b** Trendline chart of the number of glaucoma medications comparing preoperative to postoperative follow-up data.

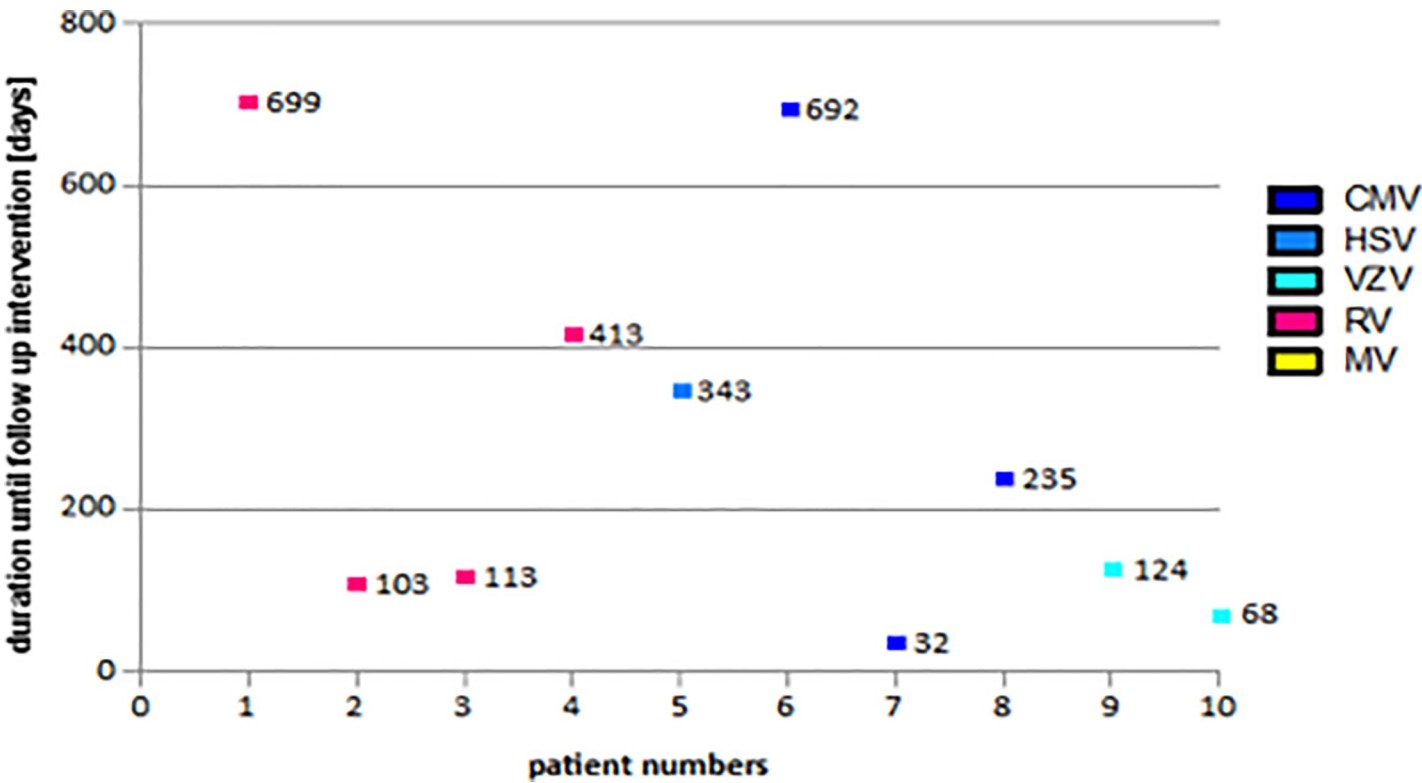

**Fig 3. Duration of follow-up interventions in glaucoma patients (CMV = cytomegalovirus, HSV = herpes simplex virus, MV = multiple virus, RV = rubella virus, VZV = varicella-zoster virus).**

with acute IOP spikes. These IOP elevations can make acute glaucoma surgery necessary or end in developing a chronic course of the disease. Our data revealed that CMV is also not as uncommon in the European population, particularly as an acute uveitis (96%) while RV takes on a chronic phenotype (84%) which we confirmed in the immune mediator analysis in our prework.[10] The clinical parameters in CMV show concordance with the reports from previous studies; laterality is almost always unilateral, and endotheliitis and/or corneal edema can manifest due to IOP elevation. In contrast, HSV AU usually follows an acute recurrent course, is typically unilateral, presents KPs, and elevated IOP. In addition, many HSV patients show a chronic course with a diagnosed keratitis, although HSV AU can also occur in the absence of corneal involvement.[19,20] Our data align with comparative research.[19,20] We also observed iris synechia and iris atrophy in some patients which is more common in HSV AU. [18,41,42] A bilateral manifestation of HSV infections is a challenging diagnosis (14% of our cohort) and the focus is on avoiding misdiagnosis as non-infectious uveitis.[43] Frequently high IOPs >30 mmHg were observed in 46 to 90% of patients while in our cohort only 4% of HSV patients were affected.[17,18,20] Only 9.6% of the glaucoma cohort were associated with HSV. The outlined research data, however, presented only few patients, limited to 8 to 39 individuals, in contrast to our patient cohorts.

Furthermore, we can present a large study cohort of VZV cases (45 patients) with confirmed VZV virus in AH. This number has not been presented in the literature to the best of our knowledge. In general, the incidence of VZV increases with age, especially patients above the age of 60 years. VZV AU is commonly associated with herpes zoster ophthalmicus with or without skin rash affecting the distribution of the ophthalmic nerve. Surprisingly, our VZV

**Table 5. Correlation of antibody index of Goldmann-Witmer coefficient and glaucoma parameters.**

| | | RV | | CMV | | HSV | | VZV | | MV | |
|---|---|---|---|---|---|---|---|---|---|---|---|
| | | AI | p | AI | p | AI | p | AI | p | AI | p |
| **IOP before AC stab incision** | IOP≤21 mmHg | 21.6 ±41.18 (N = 56) | 0.180 | 28.4 ±42.3 (N = 18) | 0.943 | 9.07 ±13.7 (N = 60) | 0.793 | 7.16±9.9 (N = 30) | 0.373 | 4.3±1.7 (N = 10) | 0.22 |
| | IOP≥22 mmHg | 45.7 ±73.8 (N = 20) | | 27.1 ±74.6 (N = 38) | | 10.2 ±13.06 (von N = 13) | | 13.6 ±26.4 (von N = 15) | | 2.98±2.7 (N = 3) | |
| **IOP before AC stab incision** | IOP≤30 mmHg | 21.3 ±42.9 (von N = 70) | 0.066 | 22.7 ±35.3 (N = 32) | 0.697 | 9.4±13.7 (N = 71) | 0.476 | 9.7±18.7 (N = 37) | 0.738 | 4.3±1.7 (N = 10) | 0.57 |
| | IOP≥31 mmHg | 104.8 ±89.3 (N = 6) | | 33.99 ±92.3 (N = 24) | | 4.56±2.6 (von N = 2) | | 7.4±7.6 (von N = 8) | | 5.98±2.7 (N = 3) | |
| **Topical glaucoma therapy** | no | 32.2 ±57.4 (N = 45) | 0.293 | 31.1 ±51.4 (N = 10) | 0.889 | 8.3±7.06 (N = 35) | 0.881 | 11.6 ±21.6 (N = 27) | 0.175 | 3.6±1.1 (N = 5) | 0.28 |
| | yes | 21.7 ±44.0 (N = 31) | | 26.7 ±68.7 (N = 46) | | 10.16 ±17.5 (von N = 38) | | 5.79 ±5.32 (von N = 18) | | 5.36±2.2 (N = 8) | |
| **Systemic glaucoma therapy (acetazolamid)** | no | 26.2 ±49.9 (N = 70) | 0.665 | 34.4 ±82.9 (N = 34) | 0.211 | 9.58 ±13.9 (N = 68) | 0.437 | 8.5±17.6 (N = 35) | 0.563 | 3.82 ±0.89 (N = 9) | 0.14 |
| | yes | 47.8 ±78.7 (N = 6) | | 16.8 ±16.02 (N = 22) | | 4.97 ±2.02 (von N = 5) | | 11.99 ±16.4 (N = 10) | | 6.64 ±2.59 (N = 4) | |
| **Glaucoma surgery** | no | 28.54 ±53.6 (N = 65) | 0.319 | 29.6 ±69.0 (N = 50) | 0.268 | 9.56 ±13.9 (N = 68) | 0.679 | 9.59 ±17.6 (von N = 43) | 0.068 | 4.51 ±1.78 (N = 11) | 0.98 |
| | yes | 24.3 ±45.7 (N = 11) | | 9.8 ±10.99 (N = 6) | | 5.23 ±2.32 (N = 5) | | 3.15 ±0.13 (N = 3) | | 5.68±3.8 (N = 2) | |

AC = anterior chamber, AI = antibody index, CMV = cytomegalovirus, HSV = herpes simplex virus,
IOP = intraocular pressure, VZV = varicella-zoster virus

cohort did not reveal any skin lesions in the past and all patients were immunocompetent. VZV patients followed an acute or chronic course in an equal distribution, laterality was in most cases unilateral, but also bilateral. Although HSV and VZV are clinically similar, the elevated IOP was more common in VZV than in HSV and the vitritis was more prominent in VZV than in HSV patients. In contrast, our RV patients showed vitreous haze in two-thirds of cases. To note, RV AU has a wider spectrum of clinical findings than the clinical features typical of FUS.[44] FUS is a clinical syndrome which is associated with RV. A recent study of Groen-Hakan et al. confirmed that RV AU and FUS are not exchangeable.[44] However, the combination of AU and vitreous haze can cause a dilemma based on suggestion of the diagnosis of intermediate uveitis. Moreover, a study showed that 98% of FUS patients demonstrated disc hyperfluorescence on fluorescence angiography.[45] Therefore, a confirmation of RV infection in AH is meaningful to ensure the correct diagnosis. Unnecessary administration of corticosteroids prevents further complications such as cataract and glaucoma development.

Interestingly, the MV cohort showed the highest mean age compared to all other cohorts and two-third were male patients. To our knowledge, a cohort with multiple tested viruses was not mentioned in current literature. This is most likely because not all viruses were tested simultaneously in comparative research. We should be aware that older patients in particular might show coinfections with significant AI titers for different and MV due to disruption of the blood-aqueous humor barrier–which might play a crucial role in therapeutic decision and visual outcome of the patients. Interestingly, no previous publication showed positive results for more than one of these viruses. Only a recent study measured a positive GWC for MV in five patients, but they were negative in PCR for all investigated agents.[44] However, it was not further discussed.

## Glaucoma

The most common and known complication of VAU is the development of secondary glaucoma.[46] Dick et al. found that uveitis caused a significantly higher 5-year risk of glaucoma (20% vs. 9%) and severe consequences for visual acuity.[47] Several studies reported that secondary glaucoma develops in 10–40% of VAU subjects during the course of the disease[16–20,46] which is in line with our data. Prevalence of secondary glaucoma was similar in RV, HSV, and VZV associated uveitis (p = 0.686).[18] CMV showed very high IOP values during active inflammatory episodes in comparison to RV, HSV, and VZV.[10,11,15,36] Exceptional cases in uveitis have always been reported and have to be considered in therapeutic decisions. In our study cohort, glaucoma surgery was necessary in approximately 10% of all VAU participants. Considering the virus antigen distribution in our study cohort, CMV and rubella virus associated uveitis were on risk for developing glaucoma and needing glaucoma surgery. In a recently published paper, analyzing CMV AU surgery of uncontrolled IOP was necessary in up to 25.7% of the study cohort.[48] In RV AU, no comparative literature considering different options in glaucoma surgery can be found. Our treatment regime of antiviral and glaucoma therapy can be seen as playing a fundamental role for the successful survival of our study cohort leading to an adequate number of cases needing glaucoma surgery. Protective factors in preventing first-line surgery are early intravenous or oral antiviral medication.[47,48]

## MIGS–only an exceptional use in VAU

Regarding different glaucoma surgeries, MIGS showed a reliable IOP reduction in the short-term period and then started to fail in 56% of all MIGS cases (9 out of 16) over the course of two years. The great advantage of MIGS can be found in its low complication level.[11,27–30] Thus, this surgical principle was promising in VAU eyes to avoid inflammatory recurrence due to the intra- and perioperative interventions needed in filtrating procedures. Glaucoma surgery, especially in young patients with IOP >30mmHG, remains challenging. Avoiding vision-threatening complications such as choroidal effusion, bleeding, and postoperative hypotony need to be the primary goal of a surgeon. Thus, different MIGS devices were used in our VAU cohort as first-step procedure after gaining knowledge of efficiency in primary open angle glaucoma (POAG) and exfoliative glaucoma. According to our retrospective data, we can state that MIGS can be used in secondary glaucoma, however, the device should be carefully chosen for the individual needs of a patient by specialized glaucoma and uveitis experts. MIGS might work (approximately 50% of our MIGS cohort) in individual secondary glaucoma cases but there are no data showing evidence for building up guidelines to generally start with MIGS. The size of our glaucoma surgery cohort (n = 27) does not allow statistically relevant conclusions about differences of MIGS devices.

There exists one comparative study showing a significant reduction of IOP 40±10 mmHg (range 33–58 mmHg to 13±1 mmHg) and glaucoma medication (decrease of 2.3 number of medications) in Trabectome® surgery in CMV AU, however, it only considers a one-year period of time.[11] Shimizu et al. reported on filtrating and Trabectome® surgery in different uveitis cohorts including VAU and found a higher survival rate of trabeculectomy (83%) compared to MIGS (75%).[16] They reported a higher risk of surgical failure in young male patients with nongranulomatous uveitis and prolonged postoperative inflammation.[16] Thus, trabeculectomy as a filtrating procedure remains the widely accepted standard in VAU to achieve an effective IOP reduction. This study is the first, to the best of our knowledge, to analyze the outcome of MIGS and filtrating procedures in different viral types of a large VAU cohort.

## High-failure rate of glaucoma surgery in VAU

In comparing the results of VAU secondary glaucoma to POAG data, we would have to report about ineffective surgical procedures considering a failure rate of 37% of the complete cohort. Iwao et al. found a significantly higher 3-year success rate of 89.7% in POAG compared to 71.3% in uveitic glaucoma.[49] In our cohort, 10 patients received a second intervention due to surgical failure and uncontrolled individual pressure (Trabectome® n = 4; iStent inject® n = 3; canaloplasties ab externo n = 2, TE+ Mitomycin C n = 1). These results imply a failure rate of 11.1% in trabeculectomy (1/9), 50% in Trabectome® (4/8), 50% in iStent inject® (3/6) and 100% in canaloplasties (2/2). These data cannot be considered as statistically significant due to low patient numbers. Instead, it should only be used to highlight a trend. Interpretation of these data does not allow conclusions about IOP reduction between single MIGS procedures. It is of great importance to choose the right MIGS device to achieve a long-lasting IOP reduction. MIGS can be recommended to start with in young patients and patients with high IOP in order to reduce intra- and perioperative risks of vision loss.

Considering the trabeculectomy surgery, our cohort showed a low-failure rate in contrast to comparative research. Kwon et al. reports a failure rate of as high as 51.9% in trabeculectomies.[50] Shimizu et al. stated data of failure in filtrating surgery and Trabectome® in 21.3% of all cases (n = 10 out of 47).[16] Additionally, a 25% failure rate of trabeculectomy was found by Ceballos et al.[51] These results align more accurately with our results regarding the filtrating surgery.

Interestingly, VZV showed a failure rate in the first three months after the first surgical intervention, while in RV and CMV AU, a high mean variation was observed. The informative value is limited by the small number of subjects. To the best of our knowledge, no comparative data of VZV and RV were found in literature. A significant risk factor of failure lies in the postoperative relapse of inflammation. Thus, a controlled inflammatory situation pre- and postoperative is crucial for therapeutic success.[16,52] In addition, hypotony is one of the most feared complications following glaucoma surgery in uveitis. In our study cohort, there were no cases of hypotony to report. Kwon et al. reported early hypotony rates of trabeculectomy and Ahmed valve implantation at 30%, as well as late hypotony at 11–15%.[50] In contrast, the research group of Iwao et al. found no significant difference in the frequency of highly feared surgical complications such as bleb leakage, hypotensive maculopathy, hemorrhage, and endophthalmitis.[49] This is in line with our data. In summary, the most important decision in minimizing failure risk is to obtain a stable intraocular inflammatory situation including antiviral and glaucoma medication before starting surgical interventions, and if possible, at intervals of at least three months of the last acute inflammation.[47] Patients, however, have to

be informed about the complication profile and close postoperative follow-up appointments in different viral VAU types. [33–35,46,49,52]

## Antibody index and immune mediators

The question of whether the severity of VAU can be measured by AI or immune mediators remains relevant. We observed a higher AI in CMV and RV in patients with IOP over 30mmHg compared to other patients with moderate IOP. We assume that there might be a relation between AI, inflammatory process, and glaucoma development.[11] Interestingly, we measured a negative correlation between mediators and IOP in CMV AU which is published in our previous work.[10] IOP elevation might be due to several parameters: a trabecular meshwork obstruction and a decreased outflow of AH, a reduction of AH drainage from the anterior chamber and degradation of mediators, or apoptosis. These data focused on all immune mediators in all groups which showed a similar distribution. Surprisingly, we observed that women showed the lowest immune mediators´ levels in the CMV cohort. Over-all, it is difficult to state a clear judgement about the immune mediators. Therefore, prospective studies need to be conducted to follow-up on these trends.

## Limitations

This study has several limitations. Its retrospective nature leads to lack of information on the clinical findings such as type and distribution of KPs. Because we are a tertiary referral center, we sometimes see the patients very late in a chronic status and the onset of the disease could not be established. After confirmation of diagnosis, some patients continued their follow-up with the first center. In contrast, the glaucoma cohort showed a close follow-up. Normally, a correct diagnosis was made by the analysis of the AH and a PCR was omitted. There was not enough AH left in all cases to perform the immune mediator analysis. Clinical data and experience have always been one of the most important features in diagnosis and therapy of uveitis. Therefore, although retrospective assessed, our data of this large number of VAU patients are still valuable information for the outpatient clinicians. Due to the retrospective study design and that data have been assessed starting in 2009, retinal nerve fiber layer (RNFL) was not analyzed or available in all patients of the study group. The significance of this information as glaucoma parameter in uveitis was contained by Moore et al.[53] Their study group reported an increased RNFL thickness than anticipated in secondary glaucoma.[53] Furthermore, we did not analyze MIGS with combined cataract surgery, thus this could be an idea to follow-up on when IOP elevation might be due to a severe cataract and inflammation.

Prospective studies, including repeated AH analysis over time, need to be conducted to evaluate a correlation between inflammation and the rise of cytokine levels in comparison to glaucoma development, progression, and necessary surgical treatment.

## Conclusion

There are several different clinical characteristics which describe the individual virus entities. In some cases, however, the clinical findings present similarities which makes a determination of the correct virus uncertain. Elderly male patients in particular could present MV simultaneously. Therefore, it is worthwhile to analyze antibody synthesis in AH to establish the appropriate treatment at an early stage of disease. More than half of our patients already had a chronic course of VAU and were not yet set with adequate therapy. During this process, cataract and glaucoma may develop as a vision-threatening complication of uveitis. In particular, the glaucoma therapy and surgical interventions remain challenging in VAU and need to be conducted by specialized centers. In our study, we were able to show a significant IOP

reduction and thus a controlled glaucoma situation by filtrating surgery. Additionally, MIGS can certainly be used as first-line treatment in individual cases of VAU, such as young age and high preoperative IOP > 30mmHG to avoid choroidal effusion and hypotony. At this time, virus association for the outcome of glaucoma surgery cannot be stated because of the low total number of patients needing surgical interventions.

## Supporting information

**S1 Table. Summary of immune mediators' levels (pg/mL) in log10.**
(DOCX)

**S1 Fig. Immune mediators' distribution in four study cohorts.**
(TIFF)

**S1 Data.**
(XLSX)

## Acknowledgments

The abstract was presented as oral presentation at Free paper session at International Ocular Inflammation Society Congress in Taiwan on November 13-16[th] 2019.

## Author Contributions

**Conceptualization:** Dominika Pohlmann.

**Data curation:** Dominika Pohlmann, Milena Pahlitzsch, Sylvia Metzner, Matthias Lenglinger.

**Formal analysis:** Dominika Pohlmann, Milena Pahlitzsch.

**Investigation:** Dominika Pohlmann, Stephan Schlickeiser.

**Methodology:** Dominika Pohlmann, Milena Pahlitzsch, Stephan Schlickeiser.

**Project administration:** Dominika Pohlmann.

**Resources:** Stephan Schlickeiser.

**Software:** Stephan Schlickeiser.

**Supervision:** Stephan Schlickeiser, Uwe Pleyer.

**Validation:** Dominika Pohlmann, Milena Pahlitzsch.

**Visualization:** Dominika Pohlmann, Milena Pahlitzsch, Anna-Karina B. Maier, Sibylle Winterhalter, Uwe Pleyer.

**Writing – original draft:** Dominika Pohlmann, Milena Pahlitzsch.

**Writing – review & editing:** Stephan Schlickeiser, Eckart Bertelmann, Sibylle Winterhalter, Uwe Pleyer.

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
