## [Decision Letter · Decision Letter 0]

16 Nov 2019

PONE-D-19-29232

Virus-associated anterior uveitis and secondary glaucoma:Diagnostics, clinical characteristics and surgical options

PLOS ONE

Dear Dr. Pohlmann,

Thank you for submitting your manuscript to PLOS ONE. After careful consideration, we feel that it has merit but does not fully meet PLOS ONE’s publication criteria as it currently stands. Therefore, we invite you to submit a revised version of the manuscript that addresses the points raised during the review process.

We would appreciate receiving your revised manuscript by Dec 31 2019 11:59PM. To enhance the reproducibility of your results, we recommend that if applicable you deposit your laboratory protocols in protocols.io, where a protocol can be assigned its own identifier (DOI) such that it can be cited independently in the future. For instructions see: http://journals.plos.org/plosone/s/submission-guidelines#loc-laboratory-protocols

We look forward to receiving your revised manuscript.

Kind regards,

Ahmed Awadein, MD, Ph.D, FRCS

Academic Editor

PLOS ONE

Journal Requirements:

2. Please include in your financial disclosure statement the name of the funders of this study (as well as grant numbers if available). At present, this information is only available in the funding section of your manuscript.

3. In the ethics statement in the manuscript and in the online submission form, please provide additional information about the patient records and samples used in your retrospective study. Specifically, please ensure that you have discussed whether all data were fully anonymized before you accessed them and/or whether the IRB or ethics committee waived the requirement for informed consent. If patients provided informed written consent to have data from their medical records used in research, please include this information.

 Dr. Dominika Pohlmann is participant in the BIH Charité Clinician Scientist Program funded by the Charité- Universitätsmedizin Berlin and the Berlin Institute of Health.

The authors are free of a financial interest.

Reviewers' comments:

Reviewer's Responses to Questions

**Comments to the Author**

1. Is the manuscript technically sound, and do the data support the conclusions?

Reviewer #1: Yes

Reviewer #2: Yes

2. Has the statistical analysis been performed appropriately and rigorously? 

Reviewer #1: Yes

Reviewer #2: I Don't Know

3. Have the authors made all data underlying the findings in their manuscript fully available?

Reviewer #1: Yes

Reviewer #2: Yes

4. Is the manuscript presented in an intelligible fashion and written in standard English?

Reviewer #1: Yes

Reviewer #2: Yes

5. Review Comments to the Author

Reviewer #1: I congratulate the authors for a well written and clear manuscript and a well designed study. The tables are very well presented although some table are rather exhaustive

The results are very informative and relevant to the clinical practice

Reviewer #2: The study question and idea: the idea is interesting and addressing an important gap in literature , the manuscript is also well written some points however still need to be addressed

1. the number of patients mentioned throughout the manuscript is 270 patients (=eyes), this is unclear as the authors mentioned in table (1) that 40 cases were bilateral

2. in table (1) : the authors mentioned that 123 case experienced an acute course of uveitis (hypertensive uveitis) , It is not understood why operation was needed during acute iridocyclitis which most cases can be temporarily controlled with medical treatment till remission occurs , even though in the rare circumstances where urgent surgery is needed , it is very difficult to judge whether the IOP has been surgically controlled or has it been the natural course of the disease . also evaluation of the success rate of surgical intervention in these cases is highly variable and can cause a statistical bias .

3. the surgical success rate was not clearly defined and the number of acute attacks could have been statistically correlated with the IOP control

4. it is not clear why the study is retrospective although the follow up plan is clearly defined in page 11 "...Patients

attended clinics preoperatively as well as follow-ups one day, six weeks, three months, six months, one

year, and two years postoperatively."

5. In table 4 : 2 cases needed CPC , this is needes further elaboartion as CPC in uveitic glaucoma is not favourable being associated with high incidence of pthisis bulbi so please outline the points of view in taking such decision

5. At which point was the "Goldmann-Witmer " test done ? during the attack of AAU or in CAU ? and how sensitive/ specific this test is

6. the conclusion "MIGS can be used as first-line-treatment in

individual cases, however, the device needs to be carefully chosen by experienced specialists based

on the individual needs of the patient. Filtrating glaucoma surgery remains the gold standard" can not be drawn as a fact from this study as it needs prospective evaluation and standardization of patients criteria and severity of glaucoma to be properly evaluated

6. PLOS authors have the option to publish the peer review history of their article (what does this mean?). If published, this will include your full peer review and any attached files.

Reviewer #1: No

Reviewer #2: Yes: Riham S.H.M. Allam

---

## [Author Response · Author response to Decision Letter 0]

29 Nov 2019

Dear Prof. Rosenbaum, 

Dear Prof. Awadein,

Dear Reviewers,

We sincerely thank you for your constructive and astute remarks on our manuscript and your thoughtful recommendations. We have taken the opportunity to carry out substantial enhancements. 

We hope now that all issues are well processed. 

We also uploaded an anonymized data set to replicate our study findings as Supporting Information files.

All authors have approved this revised version and hope that you now find it acceptable for publication. 

Kind regards,

Dominika Pohlmann, MD and Milena Pahlitzsch, MD

---

## [Editor Report · Decision Letter 1]

7 Jan 2020

PONE-D-19-29232R1

Virus-associated anterior uveitis and secondary glaucoma:

Diagnostics, clinical characteristics, and surgical options

PLOS ONE

Dear Dr. Pohlmann,

Thank you for submitting your manuscript to PLOS ONE. After careful consideration, we feel that it has merit but does not fully meet PLOS ONE’s publication criteria as it currently stands. Therefore, we invite you to submit a revised version of the manuscript that addresses the points raised during the review process.

ACADEMIC EDITOR: 

1- Please replace the word puncture (occasionally misspelled in some parts of the text as punction) with a more scholarly word like paracentesis or AC stab incision

2- Page 18 Place replace the word "positively tested" with "positive" or "positive results"

We would appreciate receiving your revised manuscript by Feb 21 2020 11:59PM. To enhance the reproducibility of your results, we recommend that if applicable you deposit your laboratory protocols in protocols.io, where a protocol can be assigned its own identifier (DOI) such that it can be cited independently in the future. For instructions see: http://journals.plos.org/plosone/s/submission-guidelines#loc-laboratory-protocols

We look forward to receiving your revised manuscript.

Kind regards,

Ahmed Awadein, MD, Ph.D, FRCS

Academic Editor

PLOS ONE

---

## [Author Response · Author response to Decision Letter 1]

8 Jan 2020

Dear Prof. Rosenbaum, 

Dear Prof. Awadein,

We sincerely thank you for your constructive remarks on our manuscript. We have taken the opportunity to carry out substantial enhancements. 

We hope now that all issues are well processed. 

All authors have approved this revised version and hope that you now find it acceptable for publication.

---

## [Editor Report · Decision Letter 2]

4 Feb 2020

Virus-associated anterior uveitis and secondary glaucoma: Diagnostics, clinical characteristics, and surgical options

PONE-D-19-29232R2

Dear Dr. Pohlmann,

We are pleased to inform you that your manuscript has been judged scientifically suitable for publication and will be formally accepted for publication once it complies with all outstanding technical requirements.

With kind regards,

Ahmed Awadein, MD, Ph.D, FRCS

Academic Editor

PLOS ONE
---

## [Editor Report · Acceptance letter]

7 Feb 2020

PONE-D-19-29232R2 

Virus-associated anterior uveitis and secondary glaucoma: Diagnostics, clinical characteristics, and surgical options 

Dear Dr. Pohlmann:

I am pleased to inform you that your manuscript has been deemed suitable for publication in PLOS ONE. Congratulations! Your manuscript is now with our production department. 

With kind regards,

on behalf of

Dr. Ahmed Awadein 

Academic Editor

PLOS ONE